

# Comparison of MODIS and VIIRS cloud properties with ARM ground-based observations over Finland

Moa K. Sporre[1], Ewan J. O'Connor[2,3], Nina Håkansson[4], Anke Thoss[4], Erik Swietlicki[1], and Tuukka Petäjä[5]

5    1 Department of Physics, Lund University, Lund, Sweden
2 Department of Meteorology, University of Reading, Reading, UK
3 Finnish Meteorological Institute, Helsinki, Finland
4 Swedish Meteorological and Hydrological Institute, Norrköping, Sweden
5 Department of Physics, University of Helsinki, Helsinki, Finland

10    *Correspondence to*: M. K. Sporre (moa.sporre@nuclear.lu.se)



**Abstract.** Cloud retrievals from the Moderate Resolution Imaging Spectroradiometer (MODIS) instruments aboard the satellites Terra and Aqua and the Visible Infrared Imaging Radiometer Suite (VIIRS) instrument aboard the Suomi-NPP satellite are evaluated using a combination of ground-based instruments providing vertical profiles of clouds. The ground-based measurements are obtained from the Atmospheric Radiation Measurement program (ARM) mobile facility, which was deployed in Hyytiälä, Finland, between February and September 2014 for the Biogenic Aerosols – Effects on Clouds and Climate (BAECC) campaign. The satellite cloud parameters cloud top height (CTH) and liquid water path (LWP) are compared with ground-based CTH obtained from a cloud mask created using lidar and radar data and LWP acquired from an a multi-channel microwave radiometer. Clouds from all altitudes in the atmosphere are investigated. The clouds are diagnosed as single or multiple layer using the ground-based cloud mask. For single layer clouds, satellites overestimated CTH by 340 m (16 %) on average. When including multilayer clouds, satellites underestimated CTH by on average 320 m (9 %). MODIS collection 6 overestimated LWP by on average 13 g m$^{-2}$ (11 %), interestingly, LWP for MODIS collection 5.1 is slightly overestimated by Aqua (4.56 %) but is underestimated by Terra (14.3 %). This underestimation may be attributed to a known issue with a drift in the reflectance bands of the MODIS instrument on Terra. This evaluation indicates that the satellite cloud parameters selected show reasonable agreement with their ground-based counterparts over Finland, with minimal influence from the large solar zenith angle experienced by the satellites in this high latitude location.

## 1 Introduction

Clouds are a very important component of the Earth's energy budget since they contribute to a large fraction of both the reflected shortwave radiation and absorbed longwave radiation. The magnitude and sign of the cloud impact depends on the cloud altitude, and a correct representation of the cloud distribution in the vertical is crucial to obtain a good estimate of the Earth's energy budget. One of the largest uncertainties in the global climate models used to predict the future climate is the representation of clouds, their feedbacks and their interaction with short and longwave radiation (Dolinar et al., 2015;IPCC, 2013). Satellite cloud retrievals provide cloud distributions on a global scale, which are used to assess global climate models (e.g. Dolinar et al., 2015). It is therefore of great importance to evaluate the satellite-retrieved cloud properties and investigate whether they provide an accurate representation of the cloud fields.

The Moderate Resolution Imaging Spectroradiometer (MODIS) instrument is carried aboard the satellites Terra and Aqua, providing information on clouds (and many other terrestrial and atmospheric properties) since 2000 and 2002, respectively. Terra and Aqua are polar-orbiting with each MODIS instrument providing an image of the whole globe every 2 days (Platnick et al., 2003). A new collection (number 6) of MODIS Level 2 products was released in 2014. Several updates to the cloud product were implemented for collection 6 (hereafter C6) compared to the previous collection 5.1 (hereafter C5.1). The cloud top properties are now provided at 1 km spatial resolution along with new products, such as cloud top height (CTH) (Baum et al., 2012). Furthermore, the cloud optical properties have been modified, including updates of the radiative transfer model and look-up tables (Platnick et al., 2014) and the thermodynamic phase retrievals (Baum et al., 2012).



The Visible Infrared Imaging Radiometer Suite (VIIRS) is carried aboard the Suomi-NPP satellite which has been in orbit since October 2011 (Cao et al., 2013). Suomi-NPP is also a polar orbiting satellite and the VIIRS sensor is similar to the MODIS sensor, but has higher spatial resolution in the infrared bands used for cloud height retrievals. Cloud products are also available from the VIIRS sensor.

In 1989 the U.S. Department of Energy initiated the Atmospheric Radiation Measurement program (ARM) with the purpose of providing ground-based measurements of clouds, and later, of aerosols and precipitation (Ackerman and Stokes, 2003). Several long-term measurement stations were implemented containing a comprehensive suite of in-situ and, passive and active remote sensing instruments. These long-term stations were later complemented with three mobile facilities and one aerial facility (Mather and Voyles, 2012).

ARM data have previously been used to validate satellite retrievals. Mace et al. (2005) evaluated cirrus retrievals from MODIS and Clouds and the Earth's Radiant Energy System (CERES) using ARM data from the Southern Great Plains Site (SGP). Data from this site were also used by Dong et al. (2008) who compared ARM low-level cloud properties with CERES-MODIS (CM) retrievals. One ARM mobile facility (AMF) was deployed in the Azores (Atlantic Ocean) for 18 months and these data were compared to CM data to validate the boundary-layer cloud retrievals (Xi et al., 2014). Other researchers have also used

ground-based measurements to evaluate MODIS (Liu et al., 2013) and CM (Yan et al., 2015) cloud property retrievals over China. MODIS cloud properties have also been evaluated with other satellite instruments such as Cloud-Aerosol Lidar with Orthogonal Polarization (CALIOP) (Holz et al., 2008) and Multi-angle Imaging Spectroradiometer (MISR) (Naud et al., 2002). This study uses ARM data from the AMF deployment in Hyytiälä, Finland, from February 2014 to September 2014 during the Biogenic Aerosols – Effects on Clouds and Climate, (BAECC) campaign (Petäjä, 2013;Petäjä et al., 2016). The AMF data is

used to evaluate the CTH from MODIS and VIIRS. The liquid water path (LWP) from MODIS C5.1 and C6 is also assessed to quantify the improvement of the updated C6 product. The investigation is not restricted to any particular cloud type but rather includes clouds from all altitudes in the atmosphere. Because the measurement site is located at a relatively high latitude, the cloud parameters can be investigated both at high and moderately high solar zenith angles (SZA). This is useful since satellite cloud retrievals have previously been found to be affected by SZA (Vant-Hull et al., 2007;Grosvenor and Wood,

2014). This study does not provide a complete validation of the MODIS cloud properties investigated, but rather provides insights into the performance of the satellite cloud retrievals. These insights can be used together with previous and future studies to improve satellite representation of cloud fields.

## 2 Method

### 2.1 Comparison Methods

Passive instruments on orbiting satellites have a much wider field of view but lower temporal resolution than most ground-based measurements. Care must, therefore, be taken when matching satellite and ground-based measurements to perform an inter-comparison at a given location. Here, one hour averaged ground-based data centred at the satellite overpass time have

been matched against satellite pixels whose centre is at maximum 15 km away from the Hyytiälä measurement station, essentially creating a circle with a diameter of 30 km around the station. Similar averaging times and areas have been used in several previous studies (Cess et al., 1996;Dong et al., 2008;Xi et al., 2014;Yan et al., 2015).

### 2.2 MODIS

The MODIS instrument is carried aboard the polar-orbiting satellites Terra and Aqua. Terra was launched in 1999, Aqua in 2002, and both satellites are sun-synchronous with Terra in a descending orbit (equatorial crossing 10:30 local solar time) and Aqua in an ascending orbit (equatorial crossing 10:30 local solar time). MODIS is a whiskbroom scanning radiometer that scans the entire Earth every two days (Platnick et al., 2003). The visible and infrared spectrum is covered by 36 bands which have spatial resolution of 250 m (2 bands), 500 m (5 bands) and 1000 m (9 bands) at nadir. The MODIS data are open access
and provided as calibrated data from the wavelength bands (level 1), instantaneous geophysical products (level 2) as well as spatially and temporally averaged geophysical products (level 3).

CTH is a new C6 level 2 cloud product produced from MODIS level 1 data. Cloud top pressure (CTP) and temperature were provided in C5.1, at 5 km spatial resolution. In C6 the spatial resolution of the cloud top properties has been increased to 1 km.

For high- and mid-level clouds, CTP is retrieved from 4 spectral bands within the 15 $\mu$m $CO_2$ absorption region using the $CO_2$ slicing method (Menzel et al., 2008). The absorption by $CO_2$ makes the atmosphere increasingly opaque at wavelengths from 13.5 to 15 $\mu$m, causing the MODIS bands in this region to be sensitive to radiances from different altitude in the atmosphere. The clear sky radiance is subtracted from the radiance in the bands in the 15 $\mu$m region and ratios of these differences are used to retrieve CTP. The method uses a top-down approach, whereby the ratio of the bands sensitive to the clouds at the highest
altitude is tested first. If this does not yield a solution, the bands sensitive to clouds at progressively lower altitudes are then tested. When a solution is found, CTH is calculated from the CTP product using gridded meteorological data from the National Centre for Environmental Prediction (NCEP) Global Forecast System.

If the $CO_2$ slicing method does not return a solution for any of the bands in the 15 $\mu$m region, CTH is derived with the InfraRed Window approach (IRW). The IRW method retrieves the cloud top temperature from the 11 $\mu$m brightness temperature (BT).
Temperature inversions in the lower atmosphere can create biases in the cloud top properties and a new technique to avoid this problem was developed for C6. Monthly average apparent 11 $\mu$m BT lapse rates are derived from collocated MODIS 11 $\mu$m BT, CALIOP cloud heights and modelled and atmospherically-corrected surface temperatures. This aims to improve the representation of the lapse rates compared to the gridded meteorological data used to obtain CTP from BT in C5.1. Hence in C6, CTP and CTH are derived from observed cloudy 11 $\mu$m BT using the monthly averaged lapse rates (Baum et al., 2012).
The other parameter compared in this study, LWP, is available in both C6 and C5.1 cloud products at 1 km spatial resolution. LWP is derived from two other cloud products, the cloud optical thickness (COT) and the effective radius ($r_e$), using the formula: $LWP = 4r_e COT/3Q(r_e)$, where $Q(r_e)$ is the extinction efficiency (King et al., 2006). This retrieval has not changed between C5.1 and C6, but modifications in the COT and $r_e$ retrievals will have a direct impact on the derived LWP. The



changes relevant to LWP in this study are: updates of the retrieval look-up tables for COT and $r_e$; the thermodynamic phase retrieval has been improved (Platnick et al., 2014); improvements to the multilayer cloud detection (Platnick et al., 2014;Wind et al., 2010). The average LWP uncertainty for the pixels used in this study is: Aqua, 21 % and Terra 23 % for C6 and Aqua, 36 % and Terra, 32 % for C5.1.

**2.3 VIIRS**

VIIRS Suomi-NPP is a scanning radiometer flying aboard Suomi-NPP, a satellite in a sun-synchronous ascending orbit crossing the equator at 13:30 local time. VIIRS has 16 M-bands with a nadir resolution of 750 m and 6 I-bands with a resolution of 375 m at nadir. It has channels both in the infrared and the visible region of the electromagnetic spectrum.

The level 1 data used in this investigation were produced at SMHI (Swedish Meteorological and Hydrological Institute) from
local reception of VIIRS data. The level 2 cloud products used here were produced with the Polar Platform System (PPS) software version 2014+patch20150327 developed by the NWC SAF (http://www.nwcsaf.org). The PPS cloud top temperature and height algorithm in PPS contains two different algorithms, one for opaque clouds and one for semi-transparent or suspected semi-transparent clouds. The 11-12 µm BT is used to determine what algorithm is used. For opaque clouds, the 11 µm BT is compared to numerical weather prediction (NWP) temperature profiles from the European Centre for Medium-Range Weather
Forecasts (ECMWF) corrected for atmospheric absorption to estimate height. A more complex histogram method is used for semi-transparent clouds, utilizing the variation across neighbouring pixels to estimate the height. The assumption that all clouds in a 32x32 square set of pixels are at the same height is used to estimate the cloud temperature. The algorithm fits a curve to the 11-12 µm BT as function of the 11 µm BT for the pixels. The cloud top temperature obtained from this fit is then compared to the model forecast temperature profile in the same manner as for opaque clouds. More algorithm details can be found in the
Algorithm Theoretical Basis Document for Cloud Top Temperature, Pressure and Height from NWC/PPS 1.1 edn., 2014 (available at http://www.nwcsaf.org).

Level 1 data were only received by SMHI until the beginning of May 2014 and hence the VIIRS data are merely available during the first 3 months of the investigation.

**2.4 AMF2 Hyytiälä**

From February to September 2014, AMF2 was deployed at the Station for Measuring Ecosystem – Atmosphere Relations (SMEAR) II station (Hari and Kulmala, 2005) in Hyytiälä, Finland. The deployment was part of a campaign called the Biogenic Aerosols Effects on Clouds and Climate (BAECC) (Petäjä, 2013;Petäjä et al., 2016), a collaboration between University of Helsinki, Finnish Meteorological Institute, University of Eastern Finland and ARM. AMF2 contains a comprehensive suite of ground-based in-situ instrumentation together with active and passive remote-sensing instruments to obtain numerous
atmospheric properties with very high temporal and spatial resolution.



### 2.4.1 Cloud Top Height

CTH is provided by the cloud mask created from a combination of the 35-GHz Ka-band ARM Zenith-pointing cloud Radar (KAZR) and the micropulse lidar or ceilometer. Gaps in the operation of the KAZR instrument were supplemented by the 95-GHz Marine W-Band ARM Cloud Radar (MWACR). The data from these instruments were processed using the Cloudnet

scheme (Illingworth et al., 2007) which diagnoses the atmospheric targets (such as aerosol, cloud, or precipitation) together with their phase if appropriate. CTH is then obtained directly as the highest cloud pixels diagnosed by this target classification. The nominal vertical and temporal resolution of CTH provided from this scheme is 30 metres and 30 seconds.

### 2.4.2 Liquid Water Path

LWP is obtained from the Radiometrics microwave radiometer, MWR, a vertically-pointing passive instrument measuring the

microwave atmospheric BT at 23.8 and 31.4 GHz. A radiative transfer model with monthly regression coefficients (Liljegren, 1999) is used to obtain column-integrated water vapour and liquid water amounts. LWP uncertainty is estimated to be ±20 g m$^{-2}$.

### 2.5 Selection Criteria

For the satellite scenes to be included in the study, the CTH retrievals had to be successful for at least 50 % of the pixels inside

a 30 km circle around the measurement station. To ensure a fair comparison, homogeneity was considered and only cases where at least 90 % of all pixels were within 1000 m of the median height were included in the analysis. If these criteria were met, the geometrical average height for the clouds within the circle was calculated as this type of averaging was most suitable. The LWP comparison was performed where satellite CTH was successfully retrieved for more than 50 % of the pixels inside the circle. Furthermore, the multilayer cloud product was used to remove pixels determined to contain several layers of clouds.

Moreover, only pixels determined to contain liquid clouds by the satellite were included in the comparison since the ground-based ARM microwave radiometer measures the LWP only for liquid clouds. Only the cases where 50 % of the pixels in the circle passed every step of the screening process were included in the LWP analysis.

For ground-based data, one hour centred on the satellite overpass time was selected for satellite overpasses that met the selection criteria above. Similar criteria for CTH homogeneity were applied: at least 90 % of the CTH values were within 1000

m of the median height; and more than 50 % of the profiles contained clouds. Again, a geometrical average CTH was calculated for cases that passed the screening. The scenes were investigated for multiple cloud layers (distinct layers separated by more than 500 m) and the fraction of multilayer clouds for each case calculated. Considering the LWP comparison, multiple cloud layers were removed from the analysis, as was performed for the satellite LWP. At least 50 % of the pixels had to contain single layer cloud only, for a geometrical average LWP to be calculated.



## 3 Results and discussion

There were 871 (Aqua) and 869 (Terra) satellite overpasses at Hyytiälä during the campaign. Of these, 322 (Aqua) and 264 (Terra) passed the selection criteria for the satellite scene CTH screening. From these scenes, 181 (Aqua) and 162 (Terra) passed the corresponding ground-based selection criteria for inclusion in the final analysis. The number of LWP cases that

passed both satellite and ground-based selection is smaller since only daytime satellite data are used. There were less data available for the VIIRS intercomparison, with a total of 300 potential Hyytiälä overpasses. 127 of these swaths passed the satellite CTH screening with 52 cases also meeting the ground-based selection criteria. There were not enough VIIRS cases that passed the LWP screening to enable an intercomparison because there were only data available during the winter months of the campaign; at high latitudes this precludes the use of visible/near IR satellite retrievals since there is not enough light.

Furthermore, during the period when there was sufficient light, most of the clouds in the VIIRS satellite scenes were classified as ice and no LWP was retrieved.

### 3.1 Cloud Top Height

The CTH retrievals are compared separately for daytime and nighttime conditions, where daytime conditions are defined as those that have a low enough SZA for the optical properties to be retrieved. For the MODIS retrievals, the maximum SZA for

optical properties is 81.4° (King et al., 2006) while the VIIRS algorithm has the maximum SZA of 72°. During February in Finland, SZA are higher than 72° at the daytime overpass and hence VIIRS data is classified as nighttime while MODIS data is classified as daytime.

The CTH intercomparison during nighttime conditions is presented in Fig. 1. For both the MODIS and VIIRS datasets, different markers are used depending on the CTH retrieval that was dominant for the case. The datasets are also divided according to

whether multilayer cloud fraction in the ARM data is smaller than 5 % (single layer case) or more than 5 % (multilayered case). Since the ARM measurements do not cover the entire satellite scene there may still be multilayered clouds present in parts of the satellite scene. The statistics in Table 1 are calculated for the single layered scenes separately and the whole dataset together. Moreover, cases where more than 50 % of the pixels are classified as ice have a cross drawn behind them (Fig. 1).

Most high-level cloud scenes contain multilayered clouds (Fig. 1). Between 39 and 57 % of the nighttime cases are classified

as containing single layer clouds (Table 1). For these, the median differences are positive between 316 and 407 m (13.0-16.3 %) indicating a satellite overestimate of CTH relative to the ground-based data. When multilayered cloud cases are included, the median difference decreases for all three datasets and become negative for Aqua and VIIRS (Table 1). There are two plausible causes for the decrease in the median differences: one is that CTH for high-level clouds are often underestimated by satellites (Holz et al., 2008) and many high cloud cases are classified as multilayer clouds; the other is that the satellite

retrievals underestimate CTH when several layers of clouds are present.

Fig. 1 also shows which retrieval method is selected for each cloud type. For MODIS, the $CO_2$ slicing method is only used on high clouds while the IRW method is used for both high- and low-level clouds. The $CO_2$ slicing method is the dominant



algorithm for 30 % of the Aqua cases but only 10 % for Terra. This difference is almost certainly due to a severe noise problem with one of the wavelength bands of the MODIS instrument aboard Terra. This band cannot be used in the $CO_2$ slicing algorithm, reducing the number of ratios available to the algorithm (from 3 to 2) for finding a successful solution. In both the Terra and Aqua datasets, there is a significant group of IRW cases for which the satellite retrievals significantly underestimate

CTH. This group contains both single and multilayer cloud cases. Previous studies (Holz et al., 2008;Naud et al., 2004), found large underestimates of CTH when the $CO_2$ slicing method does not yield a solution and the IRW method is used instead, particularly for optically thin cirrus. These large underestimates of CTH are not seen in the VIIRS data (Fig. 1c). Moreover, for VIIRS data, both CTH algorithms are used on clouds of varying altitude and there does not seem to be any particular bias by either algorithm. The colour-coding of the cases according to SZA does not show any CTH dependency on SZA at night.

The results for the daytime CTH comparison are displayed in Figure 2 and reported in Table 2. Approximately 50 % of the cases are single layer clouds for all three datasets and, similar to nighttime cases, very few high clouds are defined as single layer clouds. The Aqua (Terra) median difference between the MODIS and ground-based CTH is 358 m (241 m) for the single layered clouds only, and is reduced to -208 m (-332 m) when all cases are included; a response similar to that found for the nighttime cases. For VIIRS, the median differences are negative: -132 m for single layer clouds and -1870 m when including

multi-layer cases. For low-level clouds, CTH is very close to the 1:1 line but CTH for high cloud is underestimated by VIIRS. However, the number of daytime cases for VIIRS is low (Sect. 3), so no general conclusions regarding the performance of the algorithm should be drawn from this comparison.

The daytime results show a similar percentage to nighttime in retrieval method selection, except that now Terra and Aqua have a similar percentage of cases where the $CO_2$ slicing method is dominant. There are, as for the nighttime datasets, a few cases

where IRW retrievals significantly underestimate CTH. Moreover, there are some cases in the daytime data, where satellite CTH is several thousands of meters higher than ground-based CTH. This may occur when thin cirrus is present over optically thick low-level clouds and only detected by satellite. Optically-thin cirrus is not always detected by cloud radar and the lidar may not be able to penetrate through low-level clouds. The $CO_2$ slicing method is the dominant method for all of these cases suggesting that this method can successfully detect thin cirrus over low-level clouds. Another interesting feature in Fig. 2 is

that MODIS CTH for single layer low-level clouds seems to be somewhat overestimated at high SZA, all filled red/orange circles are above the 1:1 line (Fig. 2a and 2b).

There are to our knowledge, no prior studies evaluating MODIS C6 CTH, but previous studies have investigated the performance of the earlier collections. Collection 4 (C4) CTP has been combined with ECMWF operational analysis pressure profiles and compared to ground-based radar (Naud et al., 2005) and lidar (Naud et al., 2004) data. MODIS CTH was then

found to agree with radar CTH within 1 km for mid- and high-level clouds and within 3 km for low-level clouds (Naud et al., 2005). The comparison with the lidar showed somewhat smaller differences for the low-level clouds (-1.2 to 1.5 km) and larger differences for the high clouds (-1.4 to 2.7 km). These values are greater than the median differences between ground-based and MODIS CTH found here, but less than the extreme values. Holz et al. (2008) combined collection 5 (C5) MODIS CTP with NCEP Global Forecast System model temperature profiles and compared the calculated CTH with the satellite borne




CALIOP instrument. The MODIS retrievals were found to underestimate CTH by 1.4±2.9 km and for high clouds as much as 4 km. These differences are larger than those found in this study, and are likely due to the viewing directions relative to the clouds that ground-based and satellite active remote sensing instruments exhibit.

Both night- and daytime data were evaluated with respect to cloud fraction to determine the impact of this parameter. Cloud fraction does not appear to be associated with any specific under/overestimates or affect the magnitude of the differences. A few of the outliers do, however, have cloud fractions close to 0.5 (minimum cloud fraction, Sect. 2.5).

### 3.2 Liquid Water Path

For the satellites considered here, LWP is obtained from visible parameters and is hence only available when SZA angles are below the thresholds stated in Sect. 3.1. Here, it is investigated whether LWP from the new C6 products has a better agreement with the ground-based measurements, relative to C5.1. Figure 3 and Table 3 contain the results for all satellite scenes that passed the selection criteria while Table 4 contains the results for the satellite scenes that passed the selection criteria for both collections, i.e. can be compared directly.

As can be seen in Table 3, there were more cases selected from C5.1 than from C6 for the Aqua data; whereas, for Terra, a similar number of cases from both collections were selected. The satellites slightly overestimate LWP for C6 relative to the ground-based measurements with median differences less than 12 % (Table 3). For C5.1, Aqua marginally overestimates LWP (4.56 %), whereas Terra shows an underestimate of 14.3 %. The Terra LWP underestimate may be due to a drift in the reflectance bands of the sensor which has been corrected for in C6 (Aisheng et al., 2013). The correlation coefficients are quite high for all but the Terra C6 datasets and most cases are close to the 1:1 line when the LWP is below 200 g m$^{-2}$. For LWP values above 200 g m$^{-2}$, the scatter is large and there are more cases from the C6 dataset, likely due to the increase in the maximum value of COT (from 100 in C5.1, to 150 in C6) allowed in the retrieval. That the satellite and ground-based instruments are not viewing exactly same clouds is most likely causing some of the scatter seen in Fig. 3.

For C5.1, differences in LWP from satellite and ground-based measurements do not appear to be affected by SZA (Fig. 3). For C6 however, there does appear to some influence with respect to SZA, with a possible bias towards a satellite overestimate at high SZA. A larger dataset is necessary to confirm if this is overestimation is systematic.

There are fewer LWP cases for the direct collection intercomparison (Table 4). For Terra, 16 cases were classified as suitable in C6 but did not pass the selection criteria in C5, indicating that the changes in the algorithms made for C6 could have a significant impact. The change in the number of cases is likely a result of modifications in the cloud phase algorithm, changing how many pixels that are classified as liquid, but adjustments to the potential multilayer cloud flag and look-up tables will also affect which pixels pass the selection criteria. In general, the performance does improve when only cases where both collections pass the selection criteria (Table 4 compared to Table 3), except for the median difference in Terra C5.1 (the standard deviation does improve).

A previous study of LWP from MODIS C5 over China found that Terra and Aqua underestimated LWP by 43.3 g m$^{-2}$ and 33.6 g m$^{-2}$ compared to ground-based measurements (Liu et al., 2013). These values are larger than those found in this study




and may result from very dissimilar meteorological conditions of the investigation sites. NASA provides a product where the MODIS cloud retrievals are combined with CERES data to obtain a better understanding of the connection between longwave radiation and clouds (Minnis et al., 2011) and this too has been compared to ground-based measurements. A study over China found an overestimate of 30.2 g m$^{-2}$ for Terra CM (MODIS C5) LWP and an overestimate of 47.4 g m$^{-2}$ for the Aqua dataset

(Yan et al., 2015). Moreover, Dong et al. (2008) compared ARM measurements at a continental US site to CM data (MODIS C4) and found an overestimate by Terra (Aqua) LWP of 0.6 g m$^{-2}$ (28.1 g m$^{-2}$). Over another ARM station, at the Canary Islands, the CM (MODIS C5) LWP was underestimated by 13.5 g m$^{-2}$ (Xi et al., 2014). All three studies have only investigated overcast low-level clouds. Compared to the previous studies the differences between the MODIS and ARM LWP in this study are quite low.

**4 Conclusions**

An ARM mobile facility was deployed in Hyytiälä, Finland, from February to September 2014 as part of the BAECC campaign and provided a suitable dataset for evaluating satellite cloud retrievals at high latitudes. Ground-based measurements of CTH, obtained from lidar and radar measurements, and LWP, from microwave radiometer measurements, are compared here to three satellite instruments: the MODIS instruments aboard Terra and Aqua; and the VIIRS instrument aboard the Suomi-NPP

satellite.

There are no restrictions on CTH but the data are divided into single and multiple layers according to the cloud mask derived from the ground-based measurements. For single layer clouds, MODIS CTH is, on average, 15 % higher than ground-based measurements. For multilayer clouds, however, MODIS CTH is, on average, 7 % lower than ground-based measurements. Similar conclusions are made for the VIIRS intercomparisons during nighttime; during daytime there were not enough data to

make any general conclusions, partly a result of the high-latitude location. The MODIS IRW method frequently overestimates CTH for high-level clouds.

Single layer cloud situations only, were selected for the LWP intercomparison. Two different versions of MODIS products were evaluated, collections C6 and C5.1. The LWP for C6 shows an overestimate, relative to the ground-based measurements, of 14 % (12.5 %) for Aqua (Terra). For C5.1, there is a slight overestimate of LWP (<5 %) by the MODIS instrument aboard

Aqua, while Terra's exhibits an underestimate of about 14 %. The underestimation by Terra in C5.1 is most likely caused by a known drift in the reflectance channels, which has been corrected for in C6. Good agreement is shown between satellite and ground-based data for LWP below 200 g m$^{-2}$ but there is less agreement for LWP above this value.

The overall performance of the satellite retrievals show small median biases when compared to the ground-based observations. There are however some cloud scenes for which the satellite retrievals do not work well. Situations where thin cirrus clouds

are present over lower clouds seem to be extra problematic. This evaluation was performed at a high-latitude location to highlight any issues with large solar zenith angles, but there seemed to be little influence on the cloud parameters investigated





here. Additional evaluations of satellite cloud products performed across the globe will be necessary to draw more general conclusions regarding the performance of the investigated satellite cloud products.

**Acknowledgements**

The data from the MODIS sensors were provided by the US National Aeronautics and Space Agency through the Level 1 and

Atmosphere Archive Distribution System. The deployment of AMF2 to Hyytiälä was enabled and supported by ARM. Argonne National Laboratory's work was supported by the U.S. Department of Energy, Assistant Secretary for Environmental Management, Office of Science and Technology, under contract DE-AC02-06CH11357. The authors gratefully acknowledge the support of AMF2 (Nicki Hickmon, Michael Ritsche and others), SMEAR-II (Janne Levula and others) and the BAECC community for their support in initiating the BAECC campaign, its implementation and operation.

This work was carried out with the support of the Lund Centre for studies of Carbon Cycle and Climate Interaction, LUCCI; the European Seventh Framework Program, ACTRIS (EU INFRA-2010-1.1.16-262254), Aerosols, Clouds, and Trace gases. Research Infra Structure Network; the Strategic Research Program MERGE, Modeling the Regional and Global Earth System; the Swedish Research Council (diary no: 2010-4683). We are also grateful for the support by the Swedish Research Council and from the Nordic Council of Ministers for the Nordic Top-level Research initiative CRAICC: Cryosphere–atmosphere

interactions in a changing Arctic climate. This work was partly supported by the Office of Science (BER), U.S. Department of Energy via BAECC (Petäjä), European Commission via projects ACTRIS-TNA, ACTRIS2, BACCHUS, PEGASOS, and Academy of Finland Centre of Excellence (project number 272041).

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





Table 1. Median differences, standard deviations of the differences, correlation coefficients and p-values between MODIS and ARM cloud top heights (CTH) for single cases and all cases during nighttime.

| Satellite CTH- ARM CTH | No of samples | Median diff (m) | Std diff (m) | Median diff (%) | Std diff (%) | r | p |
|---|---|---|---|---|---|---|---|
| Aqua single | 35 | 316 | 2200 | 13.0 | 92 | 0.69 | 0.00 |
| Aqua all | 89 | -219 | 2000 | -4.66 | 42 | 0.79 | 0.00 |
| Terra single | 42 | 407 | 1800 | 15.5 | 69 | 0.61 | 0.00 |
| Terra all | 73 | 63 | 2200 | 2.29 | 78 | 0.67 | 0.00 |
| VIIRS single | 17 | 363 | 1300 | 16.3 | 57 | 0.96 | 0.00 |
| VIIRS all | 38 | -880 | 1800 | -17.8 | 36 | 0.88 | 0.00 |

Table 2. Same as in Table 1 except for daytime cases.

| Satellite CTH-ARM CTH | No of samples | Median diff (m) | Std diff (m) | Median diff (%) | Std diff (%) | r | p |
|---|---|---|---|---|---|---|---|
| Aqua single | 46 | 358 | 1800 | 19.3 | 99 | 0.35 | 0.02 |
| Aqua all | 92 | -208 | 2000 | -8.92 | 87 | 0.76 | 0.00 |
| Terra single | 44 | 241 | 1900 | 13.7 | 107 | 0.03 | 0.87 |
| Terra all | 89 | -332 | 1600 | -15.4 | 76 | 0.79 | 0.00 |
| VIIRS single | 7 | -132 | 1300 | -4.88 | 49 | 0.99 | 0.00 |
| VIIRS all | 14 | -1870 | 1400 | -26.6 | 19 | 0.93 | 0.00 |

Table 3. Median differences, standard deviations of the differences, correlation coefficients and p-values between MODIS and ARM liquid water path (LWP).

| LWP(MODIS)-LWP(ARM) | No of samples | Median diff (g m$^{-2}$) | Std diff (g m$^{-2}$) | Median diff (%) | Std diff (%) | r | p |
|---|---|---|---|---|---|---|---|
| Aqua Collection 6 | 56 | 14.0 | 60 | 12.0 | 52 | 0.75 | 0.00 |
| Terra Collection 6 | 53 | 12.5 | 110 | 10.4 | 95 | 0.53 | 0.00 |
| Aqua Collection 5.1 | 76 | 5.03 | 73 | 4.56 | 66 | 0.68 | 0.00 |
| Terra Collection 5.1 | 51 | -12.1 | 58 | -14.3 | 69 | 0.75 | 0.00 |

10 Table 4. Median differences, standard deviations of the differences, correlation coefficients and p-values between nighttime MODIS and ARM liquid water path (LWP). Only scenes approved in both the collection 5.1 and collection 6 screening are included in the table.

| LWP(MODIS)-LWP(ARM) | No of samples | Median diff (g m$^{-2}$) | Std diff (g m$^{-2}$) | Median diff (%) | Std diff (%) | r | p |
|---|---|---|---|---|---|---|---|
| Aqua Collection 6 | 50 | 12.1 | 57 | 10.6 | 50 | 0.75 | 0.00 |
| Terra Collection 6 | 37 | 3.4 | 60 | 3.15 | 56 | 0.68 | 0.00 |
| Aqua Collection 5.1 | 50 | -1.35 | 48 | -1.2 | 43 | 0.78 | 0.00 |
| Terra Collection 5.1 | 37 | -14.8 | 54 | -15.3 | 55 | 0.79 | 0.00 |





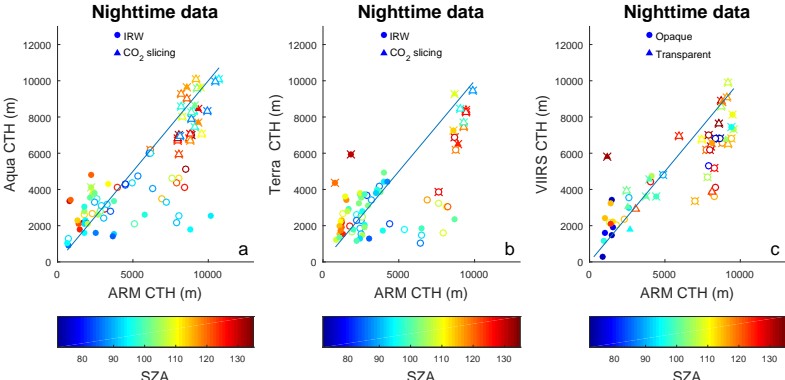

Figure 1. Scatterplots of satellite cloud top heights (CTH) versus ARM cloud top heights for nighttime cases for a) MODIS Aqua b) MODIS Terra and c) VIIRS. The colouring of the markers is according to solar zenith angle (SZA). For the MODIS plots, markers in the shape of circles indicate that the IRW retrieval of CTH was used while triangles indicate that the $CO_2$

5    slicing method has been used. For VIIRS, the triangles represent cases where the CTH was retrieved using the transparent method while the circles represent cases for which the opaque method was used. Open symbols represent cases where multiple cloud layers are present in the ARM data and the filled symbols represent single layer cases. Cases where most pixels contain ice clouds have a cross drawn behind them. The lines in the figures are 1:1 lines.

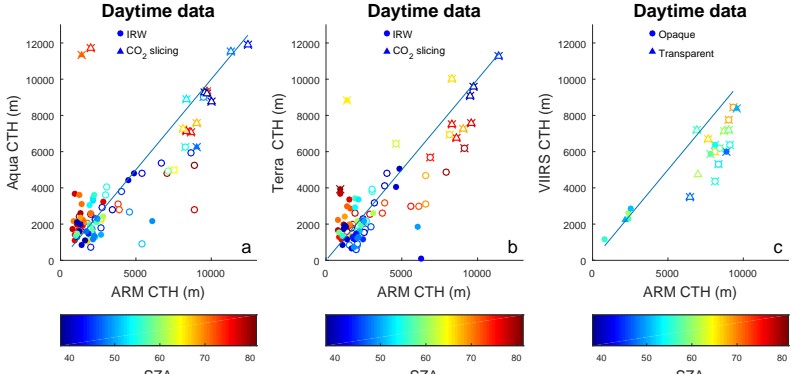

10    Figure 2. The same as Fig. 1 except for daytime cases.





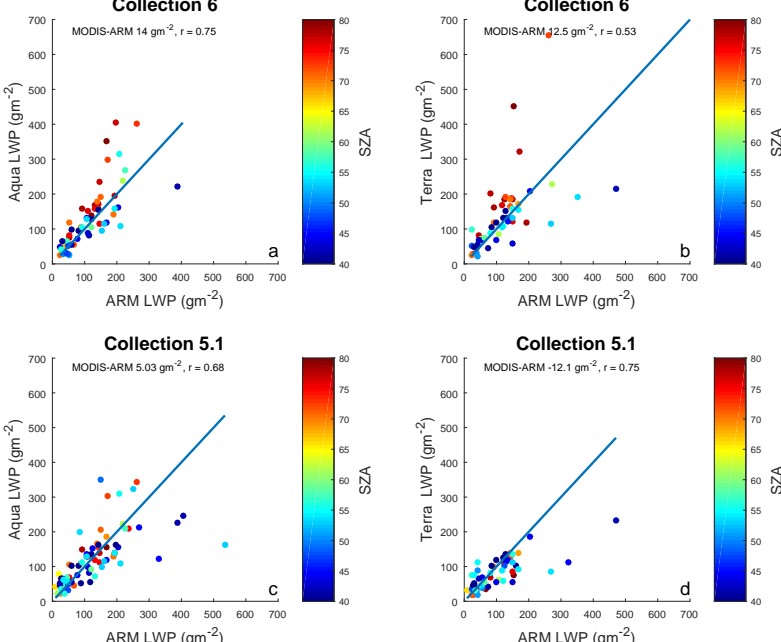

Figure 3. Scatterplots of MODIS liquid water path (LWP) versus ARM liquid water path. The two top subfigures contain data from the C6 dataset while the bottom subfigures contain data C5.1. The colouring of the circles is according to solar zenith angle. The lines in the figures are 1:1 lines.

