# Peer review of "Comparison of MODIS and VIIRS cloud properties with ARM ground-based observations over Finland"

_Atmospheric Measurement Techniques, 2016_

## Referee Comment (RC1) · Anonymous Referee #1 · 15 Mar 2016

General Comments to the Authors

The authors present a study that compares ground- and satellite-based retrievals of cloud top heights (CTH) and liquid water path (LWP). The ground-based values are derived from ARM AMF2 instrumentation deployed in Hyytiälä, Finland while the satellite retrievals are taken from Terra and Aqua MODIS Collection 6 (C6) 1-km CTHs, C5.1 and C6 1-km LWPs, and Soumi-NPP VIIRS CTHs. The results contain no surprises for those familiar with such comparisons, except perhaps dramatically smaller differences between ground- and satellite-based LWPs when contrasted to earlier studies. One of the stated goals of the study was to investigate the impact of higher solar zenith angles (SZAs) on the satellite retrievals. The authors find a weak dependence for daytime

MODIS CTH retrievals but I suspect they are really viewing zenith angle dependencies due to the high latitude location. They also find a possible SZA dependence for MODIS C6 LWP. I found the paper to be generally very well written, clear and concise. The figures and tables are appropriate and contribute greatly towards understanding of the results. I am recommending the paper be published with minor revisions that I detail in the specific comments below.

Specific Comments

Line 50: Please mention that there are also fewer bands available for CTH generation on the VIIRS, which has no 15 $\mu$m CO2 absorption bands used in the MODIS CO2 slicing method. Line 80: Awkward sentence – I would say "Therefore, care must be taken . . .". Line 89: Aqua's equator crossing time is 13:30, not 10:30. Line 97: I would add that the 5-km CTP data is still available for C6, but the 1-km has been added. I would also state that the 1-km data has been used for the study as I don't recall that it is stated explicitly. Line 101: "altitude" should be "altitudes". I would change the next sentence to read something like this: "Clear sky radiances are subtracted from observed radiances and ratios of these differences are used to retrieve CTP." And the next sentence: ". . . where the ratio of the bands sensitive to clouds at the highest altitudes are tested first." Line 105: NCEP is "National Centers for Environmental Prediction". Line 114: Mention that the apparent lapse rates are only used over ocean scenes. Land scene processing has not changed in C6. Lines 121-122: "uncertainty" should be "uncertainties" and "is:" should be "are:". Line 130: Please define NWC and SAF. Line 133: Please use and define BTD (brightness temperature difference). Line 142: Use "only" instead of "merely". Line 191: I would write, "overpasses, of which 127 passed the . . .". Line 201: Reference needed for 72° SZA value. Line 204: Please state what "dominant" means (> 50% ?). Lines 214-217: These sentences are confusing and possibly incorrect. Could you clarify these? Lines 247-249: As both MODIS CTH methods involve IR data only, I suspect this artifact is really a function of viewing zenith angle (VZA), not SZA. SZA and VZA can be correlated at high latitudes. Perhaps these data come

from higher VZAs where the cloud amounts and CTHs can be inflated with respect to near-nadir values. Line 315: I think you mean overestimated CTHs for low-level clouds and not high-level clouds. Or is it underestimating high-level clouds? Please correct.

References There are three missing references: IPCC 2013, Petäjä 2013, and Liljegren 1999

---

## Short Comment (SC1) · 29 Apr 2016

This rather straightforward analysis compares CTH and LWP from satellite retrievals to similar quantities determined from radar and microwave radiometer (MWR) from the ARM AMF package in Finland. The nice part of this study is its straightforward approach that just reports the numbers. However, it makes a few errors in interpretation and should provide a more in-depth analysis of the results. Some general and specific recommendations/comments are given below. They should be addressed before publication.

1. There is no discussion of the uncertainties or biases in the surface data set. Because the surface data set the standards, we need to know how good they are.

[Figure]

a. radar estimates of cloud top height might be pretty good for low water clouds, but how about cirrus or thick ice clouds with small crystals at cloud top. For thin cirrus, this could be answered if the surface lidar data are used to estimate cloud top. For thicker ice clouds, previous comparisons would be informative.

b. are there limitations to MWR retrievals such as effects of precipitation or thick clouds? any differences for supercooled clouds?

2. What is actually retrieved by the satellite? cloud top height or cloud radiating height? Might this influence the relationship between the radar and satellite data? Example reference: (Minnis et al. GRL, 2008)

3. A 2000-m bias might be a big deal for a low cloud at 1500 m, and not such a big deal for a cloud at 10 km. The heights should be analyzed separately for water and ice clouds.

4. The retrievals are highly sensitive to semi transparency of the clouds, especially cirrus. The results should be separated for optically thick and thin clouds (COD < 3) to provide more insight into the analysis.

Specific comments

pg. 3, line 19: "data is" should be "data are"

pg. 4, line 29: "monthly averaged lapse rates" should be "zonal monthly mean lapse rates over ocean"

pg. 6, line 4: "less" should be "fewer"

pg. 6, line 17: "is" should be "are"

pg. 6, line 27: "become" should be "becomes"

pg. 8, line 8: The VIIRS data still have some large underestimates. Need some qualification of what is meant.

pg. 8, line 17: First clause of the sentence is awkward, please rewrite

pg. 9, line 17: C6 accounted for the degradation of the Terra calibration, but not the Aqua degradation that occurred after 2008 (Doelling et al. IEEE TGRS 2015). The C6 calibrations did not account for a fundamental difference of ∼1% between Aqua and Terra that was present in C5 (Minnis et al.JAOT 2008; Dong et al. 2008). That difference will cause difference in optical depth between Terra and Aqua.

pg. 9, lines 19-20: The increase in maximum tau is unlikely to be an explanation for the difference. The maximum C6 LWP is 450 gm-2. Assuming a relatively small Re of 10 $\mu$m would correspond to COD = 67.5. Perhaps, there are larger uncertainties in the MWR data or something else going on.

pg. 10, lines 1-3: The sentence suggests that the MODIS retrievals used by CERES are the same as those used in the present comparison. They are not. The CERES MODIS retrievals were done with different algorithms, those described in the reference, Minnis et al. (2011). Please clarify.

pg. 10, lines 6-7: First, the comparisons were performed over the Azores, not the Canaries. Second, the LWP difference is 13.5 gm-2 if the larger satellite area is used, but the difference is-3.3 gm-2 if only pixels over the site are used due to island effects, which make the large area averages unrepresentative of site. Are there any systematic spatial variations over the Finland site (on coast, on a hill, etc.)?

Last paragraph, section 4: It appears that for the present LWP comparisons, the C6 results are not any better than C5, maybe even slightly worse. But they are better over this site compared to other sites. Why? Any thoughts on that?

Tables: Why are median height differences used instead of mean heights? What are the means? If you report medians, then means should also be included.

---

## Author Comment (AC1) · 28 Jun 2016

A pdf document containing the referees comments and our replies are attached. Attached is also an updated version of the manuscripts according to the reviewers' suggestions.

Please also note the supplement to this comment:
http://www.atmos-meas-tech-discuss.net/amt-2016-26/amt-2016-26-AC1-supplement.zip
* * *